# Attenuation of Spatial Memory in 5xFAD Mice by Halting Cholinesterases, Oxidative Stress and Neuroinflammation Using a Cyclopentanone Derivative

**DOI:** 10.3390/ph13100318

**Published:** 2020-10-19

**Authors:** Rahim Ullah, Gowhar Ali, Nisar Ahmad, Muhammad Akram, Geeta Kumari, Muhammad Usman Amin, Muhammad Naveed Umar

**Affiliations:** 1Department of Pharmacy, University of Peshawar, Peshawar 25120, Pakistan; 2Department of Pharmacy, National University of Pakistan, Pasrur Road, Sialkot 51310, Punjab, Pakistan; nisarahmadsatal@yahoo.com; 3Department of Pharmacology, Faculty of Pharmacy, University of Sindh, Jamshoro 76080, Pakistan; akramkhatrisu@yahoo.com (M.A.); geeta.kumari@usindh.edu.pk (G.K.); 4Department of Pharmacy, Abasyn University, Ring Road, Peshawar 25120, Pakistan; usman.amin@abasyn.edu.pk; 5Department of Chemistry, University of Malakand, Chakdara 18000, Dir (L), KPK, Pakistan; m.naveedumar@uom.edu.pk

**Keywords:** Alzheimer disease, cholinesterases, oxidative stress, neuroinflammation, spatial memory, 5xFAD mouse model

## Abstract

Alzheimer’s disease (AD) is an irreversible and chronic neurological disorder that gradually destroys memory and thinking skills. The research study was designed to investigate the underlying molecular signaling involved in the neuroprotective effects of cyclopentanone derivative i.e., 2-(hydroxyl-(3-nitrophenyl)methyl)cyclopentanone (3NCP) as a therapeutic agent for AD. In this study, In vivo studies were carried out on a well-known 5xFAD mice model using different behavioural test models such as open field, rotarod, Morris water maze (MWM), and Y-maze tests. Furthermore, in vitro cholinesterase inhibition activity assays were carried out. The frontal cortex (FC) and hippocampus (HC) homogenates were tested for the levels/activities of cholinesterases, glutathione (GSH), glutathione S-transferase (GST), and catalase. Furthermore, the hippocampal expression of inflammatory cytokines was observed via RT-PCR and western blot. The results of in vivo studies show an enhancement in the learning behavior. The 3NCP treatment reduced latency time in MWM and Y-maze tests, also increase spontaneous alternation indicate significant effect of 3NCP on memory. Furthermore, open field and rotarod studies revealed that 3NCP does not cause motor coordination deficit. The results of the in vitro studies revealed that the IC_50_ values of the 3NCP against acetylcholinesterase (*AChE*) and butyrylcholinesterase (*BChE*) were 16.17 and 20.51 µg/mL, respectively. This decline in *AChE* and *BChE* was further supported by ex vivo studies. Further, the 3NCP mitigates the GSH level, GST, and catalase activities in HC and FC. The mRNA and protein expression of inflammatory cytokines (IL-1β, IL-6, TNF-α) markedly declined in RT-PCR and western blotting. The results of the current study conclusively demonstrate that 3NCP reduces oxidative stress and mitigates neuroinflammation in 5xFAD mice, implying that 3NCP may be a potential therapeutic candidate for AD treatment in the future.

## 1. Introduction

Alzheimer’s disease (AD) is a progressive and chronic neurodegenerative disorder of the brain that gradually destroys the memory of patients and their capability of making decisions, learning, communication, and performing routine activities. Initially, in AD, short term memory gets disturbed, due to degeneration and dysfunction of neurons in the amygdala and hippocampus, progressing further to neuronal death in other cortical regions of the brain [1]. These brain regions are greatly involved in learning and memory processes. AD has been considered a major cause of dementia that represents a major socioeconomic burden for aging-societies. Hence, researchers are investigating AD pathology to develop effective therapeutic strategies. Many potential pathophysiologic mechanisms for the development of AD have been proposed, like neuroinflammation, oxidative stress, and microglial dysfunctions, among others [2,3]. This disorder is majorly linked with a cholinergic-deficit in the brain. Acetylcholinesterase (*AChE*) and butyrylcholinesterase (*BChE*) inhibit *ACh*, which plays a vital role in the AD pathogenesis. Currently, only two classes of drugs, acetylcholinesterase inhibitors (AChEIs) and *N*-methyl-d-aspartate (NMDA) receptor antagonists are approved, that are involved in the symptomatic treatment of AD. Synthetic drugs, however, are useful for the management of such diseases, however they carry severe side effects [4]. The presence of untoward effects not only limits their use in clinical setups, but also drives the modern day researcher in the search for novel targeted drug molecules with higher efficacy and lesser side effects [5]. Reactive oxygen species (ROS) produced in the human body due to complex redox reactions contribute to cellular aging and neuronal damage. The destructive effects of these ROS are diminished by certain proteins and enzymes in human body such as GSH, GST, and catalase. A decline in levels/activities of these protein and enzymes from that of the free radicals leads to oxidative stress and finally certain chronic diseases, including AD. Hence, the use of antioxidants reduces AD progression and diminishes neuronal degeneration [6].

Inflammation is another hallmark of AD. In AD brains, Aβ plaques are surrounded by activated microglial cells and reactive astrocytes. In addition, several inflammatory mediators including interleukin-1β (IL-1β), interleukin-6 (IL-6) and tumor necrosis factor-α (TNF-α) have been found within plaques [7]. Unfortunately, an effective treatment for AD is still lacking, therefore, the search for more effective anti-AD synthetic compounds with anticholinesterase, antioxidant, and anti-inflammatory properties, is desired.

Cyclopentanone derivatives have been reported to exhibit anti-inflammatory, analgesic, antioxidant, anticoagulant, anti-*AChE*, and anti-*BChE* activities [8,9,10,11]. Following up on this perspective the cyclopentanone derivative 2-(hydroxy-(3-nitrophenyl)methyl)cyclopentanone (3NCP, Figure 1) [12] was evaluated as a possible potential candidate against AD. The inhibition of *AChE* and *BChE*, also the antioxidant activities has been attributed to the presence of the nitrophenyl group [13,14,15,16,17]. The ketone moiety exhibited dose-dependent antioxidant effects, which have been considered as a possible explanation for the decline of chronic neurodegenerative disorders like AD [18].

## 2. Results

### 2.1. Behavioral Studies

#### 2.1.1. Open Field Test

This test is used to measure the locomotor activity, anxiety, and exploratory behavior in mice. Rearing, grooming behavior, and time spent in center were observed. 5xFAD mice displayed reduced number of rearing, more grooming and similar locomotor activity as compared to WT mice. While 3NCP (40 mg/kg) treated 5xFAD mice displayed a significant increase in the number of rearing (*p* < 0.001) and decrease in the number of grooming (*p* < 0.05), while, 3NCP did not change locomotor activity (Figure 2).

#### 2.1.2. Effect of Cyclopentanone Derivative “3NCP” in the Rotarod Test

In the rotarod test the 5xFAD mice and non-transgenic mice fell from the rod in same time and did not display any defects in their motor-coordination and balance. A similar latency time was observed in case of 3NCP- and galanthamine-treated 5xFAD mice (Figure 3).

#### 2.1.3. Effect of Cyclopentanone Derivative “3NCP” in the Morris Water Maze Test

In the MWM test, escape latency time was observed for each group. The results showed that the 5xFAD animals exhibited an increase in latency time as compared to WT-animals (*p* < 0.001). The 3NCP at the dose of 10 mg/kg (*p* < 0.05) and at 20 mg/kg (*p* < 0.01, *p* < 0.001) reduced latency time significantly on day 4 and 5, while at high dose of 40 mg/kg and galanthamine (8 mg/kg) caused a significant decline in latency-time on day 3 to 5, time = (F (4, 90) = 49.9, *p* < 0.0001), treatment = (F (5, 90) = 40.6, *p* < 0.0001), interaction = (F (20, 90) = 1.53, *p* < 0.09090)]. In the probe test, 5xFAD mice spent less time in the target quadrant and less number of target quadrant crossings, while 3NCP-treated mice demonstrated a reciprocal behaviour to this (*p* < 0.05, *p* < 0.01, *p* < 0.001) (Figure 4).

#### 2.1.4. Effect of Cyclopentanone Derivative “3NCP”in the Y-Maze Test

Spontaneous alternation testing derived from SAB is a behavioral assessment method. It is used to investigate exploratory behavior and cognitive function related to spatial learning and working memory. The alternation behavior of mice was determined from successive entries into three arms on overlapping triplet sets in which three different arms are entered. 5xFAD mice treated with vehicle displayed a decreased in %SAB, while treatment with 3NCP and galanthamine significantly increased the %SAB as compared to WT mice (*p* < 0.05, *p* < 0.01, *p* < 0.001), indicating that 3NCP ameliorated memory dysfunction in 5xFAD mice (Figure 4).

### 2.2. In Vitro Cholinesterases Inhibition by Cyclopentanone Derivative “3NCP”

The in vitro assay results of 3NCP against cholinesterases (*AChE*, *BChE*) are shown in Table 1. 

### 2.3. Ex Vivo Cholinesterases

#### 2.3.1. Effect of Cyclopentanone Derivative 3NCP on Cortical and Hippocampal *AChE*, *BChE* Activities

A significant decline in the percent *AChE* activities in the hippocampus (HC) was observed after treatment with 3NCP and galanthamine (F (5, 12) = 25, *p* < 0.0001) and in frontal-cortex (FC) (F (5, 12) = 34.2, *p* < 0.0001) (Figure 5A). Similarly, 3NCP and galanthamine induced a decline in the percent *BChE* activities in HC (F (5, 12) = 23.2, *p* < 0.0001) and in FC (F (5, 12) = 35.1, *p* < 0.0001) (Figure 5B). 5xFAD mice exhibited a significant enhancement in *AChE* and *BChE* activities in HC and FC (*p* < 0.001), while, 5xFAD mice treated with galanthamine and 3NCP (10, 20 and 40 mg/kg) displayed a significant decline in *AChE* and *BChE* activities (*p* < 0.05, *p* < 0.01, *p* < 0.001).

#### 2.3.2. Effect of Cyclopentanone Derivative 3NCP on GSH Level, GST, and Catalase Activities

The GSH level, GST and catalase activities declined in the HC and FC of 5xFAD transgenic mice as compared to non-transgenic WT mice. The 3NCP-treated 5xFAD mice displayed an increased level of GSH in HC (F (5, 12) = 52, *p* < 0.0001 and in FC (F (5, 12) = 28.4, *p* < 0.0001. The 3NCP at doses of 20 and 40 mg/kg enhanced the percent GSH level significantly in HC and FC (*p* < 0.05, *p* < 0.01, *p* < 0.001). The GST activity increased in 3NCP-treated 5xFAD mice in HC (F (5, 12) = 22.4, *p* < 0.0001) and in FC (F (5, 12) = 20, *p* < 0.000), at doses of 10, 20, and 40 mg/kg. Likewise, the 3NCP-treated transgenic mice displayed an increase in catalase activity (%) in HC (F (5, 12) = 18.2, *p* < 0.0001) and in FC (F (5, 12) = 14.7, *p* < 0.000) at doses of 10, 20, and 40 mg/kg (Figure 6).

#### 2.3.3. Effect of Cyclopentanone Derivative 3NCP on Cytokines (IL-1β, IL-6, TNF-α) in HC and FC Tissues

To assess the anti-inflammatory activities of 3NCP, the expression of different cytokines was evaluated in the hippocampal and frontal cortex tissues through RT-PCR. The 5xFAD mice treated with 3NCP and galanthamine revealed a significant decline in the mRNA levels of different cytokines (IL-1β, IL-6, TNF-α) as compared to the 5xFAD mice (*p* < 0.01, *p* < 0.001) (Figure 7).

#### 2.3.4. Effect of Cyclopentanone Derivative 3NCP on Cytokines (IL-1β, IL-6, TNF-α) Protein Expression in FC and HC Tissues

The western blotting results reveal that the 3NCP at dose of 40 mg/kg cause a significant down regulation ofIL-1β, IL-6, and TNF-α proteins expression in FC and HC of 5xFAD mice as compared to the vehicle treated 5xFAD mice (*p* < 0.01, *p* < 0.001) (Figure 8).

## 3. Discussion

Neuro-inflammation and oxidative stress have been described as possible pathophysiological mechanisms underlying AD. In addition, the cholinergic hypothesis contends that in AD patients, memory impairment is also linked with the deficit of cholinergic function in the brain. Although a number of drugs have been approved for the treatment of AD, a majority of these drugs have a number of undesirable side effects and yield relatively diffident benefits [19].

Acetylcholine (*ACh*) is the most crucial neurotransmitter involved in the regulation of cognitive functions. *AChE* inhibitors promote the endogenous levels of *ACh* in the brain and thus enhance cholinergic transmission, whereas *BChE* inhibitors are believed to diminish neuritic plaques in the brain [20,21]. In this study, the anti-cholinesterase activities of 3NCP were assayed both in vitro and ex vivo. The in vitro anti-cholinesterases investigation revealed that the 3NCP has concentration-dependent *AChE* and *BChE* inhibition potential. The 3NCP exhibited 84.9% and 86.5% inhibition of *AChE* and *BChE* at concentration of 1000 µg/mL, with IC_50_ values of 16.17 and 20.51 µg/mL against *AChE* and *BChE* (Table 1). The 3NCP inhibits the *AChE* and *BChE* activities in the HC and FC of the 5xFAD mice brain (Figure 2). Therefore, due to the inhibition of these enzymes, the acetylcholine levels are maintained in the synaptic cleft for a long time period, which causes stimulation of cholinergic receptors. This hike of cholinergic conduction may be the possible way in the restoration of memory in AD.

Due to its high oxygen consumption, the central nervous system is highly vulnerable to free radical and oxidative damage that play a pivotal role in the AD pathophysiology [22]. This leads to the generation of reactive oxygen species (ROS) such as superoxide anion radical, hydrogen peroxide, hydroxyl radical, and peroxyl radicals consequently resulting in oxidative stress [23,24].

Oxidative stress is known to play an important role in the pathogenesis of AD. The increased levels of Aβ1-40 and Aβ1-42 are due to more generation of oxidative products in the cortical and hippocampal tissues of AD patients [25]. Molecules with active antioxidant potential are used to scavenge such harmful radicals [19]. Glutathione, glutathione-S-transferase, and catalase are antioxidants that play a crucial role in the prevention or mitigation of free radicals mediated oxidative stress progression. Activities of these antioxidants decreases in AD individuals [26]. To remove peroxides, glutathione (GSH) works in conjunction with glutathione peroxidase (GPx) and produces oxidized glutathione (GSSG), which is reconverted to GSH by glutathione reductase (GR) with consumption of NADPH. A reduction in GSH may diminish clearance of hydrogen peroxide (H_2_O_2_) and promote formation of OH, hence increasing the free radical load, which triggers oxidative stress [1,2]. GST and catalase are antioxidant enzymes; being first line defense against oxidative injury [3]. The 3NCP has promising antioxidant potential, since it significantly improved activities of GST and catalase, and enhanced the level of GSH in HC and FC of transgenic 5xFAD mice.

Neuroinflammatory processes are fundamental characteristics of AD in which microglia are over activated, leading to increased production of pro-inflammatory cytokines. Moreover, deficiencies in the anti-inflammatory system may also contribute to neuro-inflammation [27]. It has been reported that inordinate production of inflammatory mediators and cytokines from activated microglia leads to unrestrained inflammation in neurodegenerative diseases [28,29,30]. Pro-inflammatory cytokines IL-1β, TNF-α, and IL-6 are upregulated in 5xFAD transgenic mice and AD patients [31]. TNF-α level increases in hippocampal dependent cognitive impairment in rodents [32]. Halting generation of cytokines and pro-inflammatory mediators has become a therapeutic target in managing neurodegenerative diseases, and thus their down-regulation might aid in stopping or retarding the onset of AD [33]. The RT-PCR and western blotting studies reveal that the 3NCP significantly reduced the mRNA expression and protein level of these cytokines in the hippocampus and frontal cortex of 5xFAD transgenic mice. The results from RT-PCR and western blotting envisage the redemption of brain cells from these pro-inflammatory mediators by 3NCP, advocate a role of the cyclopentanone derivatives as potential anti-inflammatory agents, and hence their role in the treatment of AD [34].

Open field (OF), rotarod, MWM, and Y-maze tests were conducted for determining the effect of 3NCP on the behavior of the test animals in terms of locomotion, motor coordination, hippocampal-dependent spatial learning and memory, and spatial recognition memory, respectively [35,36,37].

The OF assay has been an effective tool in measuring the anxiety-like behavior as reported in literature [38,39], and it has been extensively employed in the evaluation of the anxiolytic potential of many drugs, such as benzodiazepines (BDZs) [40,41,42]. A number of variables can be recorded from the OF test for the evaluation of anxiety (less distance travelling, less time spent in center and more time spent in periphery; indicate higher anxiety levels), risk assessment (sniffing, stretched attended posture; indicate inquisitive behaviour), locomotion (total distance travelled or total entries; indicate well-being) [43].

In this test, the 5xFAD mice spent same amount of time in the center as that of WT mice, showing a non-anxious behavior, showing a lack of anxiogenic effect due to 3NCP. The more time spent in the center indicates a higher degree of anxiolysis and vice versa [44].

Assessing the motor coordination and balance can be taken advantage of not only to appraise the effects of test compounds or other experimental procedures on rodents, but also to portray the motor phenotype of transgenic/knockout animals [45].

For the assessment of motor coordination, mice were tested on a rotarod at a pre-set speed (0 to 40 rpm) for 5 min [37]. The results showed no significant difference in latency times of falling among 3NCP treated and the control animals implying that the prior one did not cause any motor deficits.

The water maze test has been most widely used to explore specific facets of spatial memory. This test is based upon the assumption that animals have acquired an optimal approach to delve into their environment and escape from the water with minimum efforts (swim the shortest possible distance) [46].

In MWM assay, during training trials, spatial-information acquisition was determined and in probe trails memory retention was assessed. Results of the study suggest that 5xFAD transgenic mice exhibited cognitive decline and required more time to find the escape platform in training trials as compared to WT mice. While in probe trial, 5xFAD mice displayed a significant decrease in time spent on target quadrant and number of target quadrant crossing. The 3NCP treatment significantly decreased latency time in training trials. In the Probe trial, 3NCP treated mice spent more time in the target quadrant and also crossed the platform position more frequently than transgenic mice treated with vehicle. 

The MWM has proven to be a robust and reliable test that is strongly correlated with hippocampal synaptic plasticity and NMDA receptor function [47].

Spatial recognition and short term memory in mice can be assessed using Y-maze apparatus and this task is mediated via prefrontal cortex and hippocampal controls, respectively [48,49]. In the Y-maze test, reduced SAB (%) was observed in the 5xFAD mice, which indicates impaired spatial memory. The 3NCP treatment improved the percent spontaneous alternation, reducing spatial memory impairment. This improvement in memory demonstrates the neuroprotective potential of 3NCP against memory impairment in the 5xFAD mice.

## 4. Materials and Methods

### 4.1. Materials

The 2-(hydroxyl-(3-nitrophenyl)methyl)cyclopentanone was gifted by Dr. Muhammad Naveed Umar (Department of Chemistry, University of Malakand, Chakdara, Pakistan). Acetylcholinesterase, butyrylcholinesterase, acetylthiocholine iodide, butyrylthiocholine iodide, 5,5-dithiobis(2-nitro-benzoic)acid, galantaminehydrobromide,1,1-diphenyl-2-picrylhydrazyl, hydrogen peroxide, trichloroacetic acid, sodium citrate, glutathione-S-transferase, l-chloro-2,4-dinitrobenzene, dipotassium hydrogen phosphate, potassium dihydrogen phosphate, potassium hydroxide, ethylenediaminetetraacetic acid, boric acid, agarose, and ethidium bromide were obtained from Sigma-Aldrich (St. Louis, MO, USA). DNA extraction kit (Novel Genomic DNA Mini Kit), TRI-reagent (Bioshop, Burlington, ON, Canada), Tris (Scharlu, Barcelona, Spain), cDNA synthesis kit (ABM, Milton, ON, Canada), PCR primers (Macrogen, Seoul, Korea), PCR Master Mix, Taq polymerase (Thermo Fisher Scientific, Waltham, MA, USA), DNA Ladder, dNTPs, magnesium chloride (Invitrogen, Carlsbad, CA, USA) were purchased from local suppliers.

### 4.2. Animals

The 5xFAD mouse is a notable rodent model frequently used in the preclinical AD research. These mice present the common AD pathologies at an early age, beginning at 5 to 6 months of age [50]. They also resemble the human AD pathology in several ways, including, but not limited to, brain pathology, behavior, and biomarkers.

5xFAD mice were obtained from the Jackson Lab (Bar Harbor, ME, USA) and were housed under standard laboratory conditions in the animal house at the Department of Pharmacy, University of Peshawar, Pakistan. 5xFAD mice of either sex (age: 5–6 months) were used as a disease group and wild type mice littermates were used as control group. Experiments on animals and related procedures were approved by departmental ethical committee vide a reference number 12/EC-17/Pharm and were performed in agreement to the rules and regulations of Animals’ Scientific Procedures Act (UK), 1986.

### 4.3. In Vivo Activities

#### 4.3.1. Genotyping of Transgenic Mice

The genotyping methodology specified by Jackson Lab for strain transgenic mice was followed with minor modifications. Extraction of DNA was carried out using DNA extraction kit according to the manufacturer’s protocol. PCR studies were performed following proper cyclic conditions. Two sets of primers were used in genotyping; sequences of these primers are shown in the Table 2. PCR products were quantified through gel electrophoresis and amplified products were visualized using UV transilluminator [51].

#### 4.3.2. Behavioral Activities

##### Animal Grouping and Route of Drug Administration

Experimental mice were divided into six groups (*n* = 10 each).
Group I: WT miceGroup II: 5xFAD miceGroup III: 5xFAD-GLN (8 mg/kg, i.p)Group IV: 5xFAD-3NCP (10 mg/kg i.p)Group V: 5xFAD-3NCP (20 mg/kg i.p)Group VI: 5xFAD-3NCP (40 mg/kg i.p)

Galantamine and 3NCP were dissolved in vehicle containing normal saline, Tween-80, and DMSO in a ratio of 93:2:5. Animals were treated with vehicle/galantamine/3NCP for 28 days.

##### Open Field Test

The open field test was performed for assessment of exploratory and locomotor activities of 5xFAD mice. Animals were placed in the center of arena and allowed for 5 min to explore. The number of rearing, grooming, and square crossing, the time spent in the center were recorded [52].

##### Rotarod Test

Rotarod tests were carried out for the assessment of motor coordination deficits, following the standard method with slight modifications [53]. All mice were trained for two days, four trials each day. It was performed for 5 min by placing each animal on the accelerating rod (4–40 rpm). On the next day after training trails, testing trials were carried out and after one hour of drug administration, the falling latency time was noted.

##### Morris Water Maze

The effect of 3NCP on the spatial learning and memory of 5xFAD mice was assessed through the Morris Water Maze apparatus (MWM, diameter: 110 cm, height: 40 cm) containing opaque water (depth: 20 cm) maintained at 22–23 °C) according to reported method with minor modification [3]. The maze consisted of four quadrants with a submerged platform (diameter: 7 cm) set 1 cm below the water-surface in one of these four quadrants. All mice were trained for five days continuously (4 trials/day). The escape latency time to find the hidden platform was calculated. Those mice which failed to find the hidden platform in 60 s, were manually guided, and allowed to stay for 10 s on the platform. At the 24th h after day 5, the mice were assessed in probe trial for memory retention. The probe test was conducted without platform for 60 s. The time spent by mice in target quadrant and the number of platform place crossings were measured. All activities were video recorded using an overhead camera.

##### Y Maze Test

The Y-maze test is utilized for the assessment of short term memory. Briefly, the mice were placed randomly at the center of the Y-maze and allowed for 8 min to move freely. The sequence of arm-entries was observed visually [54]. Spontaneous alternation behavior in percent was calculated through the following formula:$$\mathrm{Spontaneous}\;\mathrm{Alternation}\;\mathrm{behavior}\;\left(\%\right)=\frac{\mathrm{Number}\;\mathrm{of}\;\mathrm{overlaping}\;\mathrm{entry}\;\mathrm{sequences}}{\mathrm{Total}\;\mathrm{arm}\;\mathrm{entries}-2}\times 100$$

### 4.4. In Vitro Assays

#### Acetylcholinesterase and Butyrylcholinesterase Inhibition Assays

Acetylcholinesterase and butyrylcholinesterase assays were performed following Ellman’s assay method [51]. Acetylthiocholine iodide (ATchI) and butyrylthiocholine iodide (BTchI) were used as substrates; degradation of these substrates leads to the formation of 5-thio-2-nitrobenzoate, which reacts with 5,5′-dithiobis (2-nitro-benzoate) (DTNB) and forms a yellow color complex, which was assayed at a wavelength of 412 nm through microplate reader.

The rate of absorbance (V = ΔAB/Δt) showed the (%) enzyme activity and inhibition by test/control samples and was calculated as follows:Enzyme inhibition (%) = 100 − enzyme activity (%)Enzyme activity (%) = 100 × V/V_max_
where V_max_ shows the maximum enzyme activity in the absence of the inhibitory agent.

### 4.5. Ex Vivo Assays

#### 4.5.1. Assessment of Inhibition of Cholinesterases (*AChE* and *BChE*) in FC and HC 

This test was performed as described elsewhere [51,55]. After behavioral experiments, all mice were killed by decapitation under ether anesthesia; HC and FC were dissected and homogenized in ice-cold 0.1 M PBS buffer (pH 8.0), the supernatant was separated and used in this assay. After *ACh* and *BCh* degradation, thiocholine and acetate-thiocholine were formed, which was combined with DNTB that produced a yellow color complex, absorbance of that complex was measured at 412 nm in the 96-well microplate reader.

#### 4.5.2. Glutathione (GSH), Glutathione S-Transferase (GST), and Catalase Assay

The level/activities of GSH, GST, and catalase in the hippocampus were determined as reported earlier in literature [25,56]. The supernatant of hippocampal homogenate was used in this test, which was obtained by homogenization and centrifugation of hippocampal tissue at speed of 1000× *g* at 4 °C for 15 min. The GSH level, the GST, and catalase activities were measured by observing the absorbance variation through microplate reader at specific wavelengths (420, 240, and 340 nm).

#### 4.5.3. RT-PCR

In RT-PCR, extraction of total RNA from the hippocampus and frontal cortex using TRI-reagent was carried out following the manufacturer’s protocol. The purity of RNA was evaluated through a UV spectrophotometer. The cDNA was synthesized from total RNA via a cDNA synthesis kit. The targeted gene (IL-1β, IL-6, TNF-α) expressions were determined (Table 2). The housekeeping gene GAPDH was used as internal control. Through agarose gel (1.5%) amplified products were separated and visualized using a UV transilluminator. The annealing temperature was 55 °C of GAPDH and IL-1β, 60 °C of IL-6 and 62 °C of TNF-α. The expression of these cytokines was calculated in the arbitrary units (A.U.s) [57].

#### 4.5.4. Western Blot Analysis

The western blot analysis was performed as reported previously [58]. Briefly, brain homogenates were quantified using Bio-Rad protein assay solution. The homogenates (30 µg protein) were fractionated using SDS-PAGE on a 15% Bio-Rad Tetra cell (BioRad, Hercules, CA, USA). After transfer, the membranes were blocked in 5% skim milk (or BSA), incubated overnight at 4 °C with primary antibodies, and cross reacting proteins were detected using ECL after reaction with horseradish peroxidase-conjugated secondary antibodies. The primary antibodies, including mouse-derived anti-IL 6 (24 kDa), anti-TNF-α (17.3 kDa), anti-p-IL-1β (17.5 kDA), and mouse-derived anti-β-actin (42 kDa) were purchased from Santa Cruz Biotechnology (Santa Cruz, CA, USA). After using membrane-derived secondary antibodies, ECL detection reagent (Bio-Rad) was used for visualization according to the manufacturer’s instructions. The densitometry analysis of the bands was performed using ImageJ software (ImageJ-win64 1.8). The density values were calculated in arbitrary units (A.U.s) relative to the untreated control.

### 4.6. Statistical Analysis

The in vitro, SWM and Y-maze tests data were analyzed through two-way ANOVA followed by Bonferroni test, while ex vivo assays and SAB tests data were analyzed by one-way ANOVA followed by Tukey’s post hoc test. Data of RT-PCR, Open field, and Rotarod tests were analyzed via one-way ANOVA followed by Dunnett *posthoc* test. Analyses were performed through GraphPad Prism 5 (GraphPad Prism Software Inc., San Diego, CA, USA) and statistical significance was set at *p* < 0.05.

## 5. Conclusions

In conclusion, the results from this study indicate beneficial effects of 3NCP in the 5xFAD mice model. Moreover, improvements in cognitive performance with integral changes in behavioral performance, cholinesterases, oxidative stress, and inflammation were found to have a prospective association in the melioration of these parameters in 5xFAD mice. Further exploration of the relationship between inflammatory cytokine polymorphisms and AD risk may add to our understanding of AD pathogenesis and conduce to improved treatment strategies.

## Figures and Tables

**Figure 1 pharmaceuticals-13-00318-f001:**
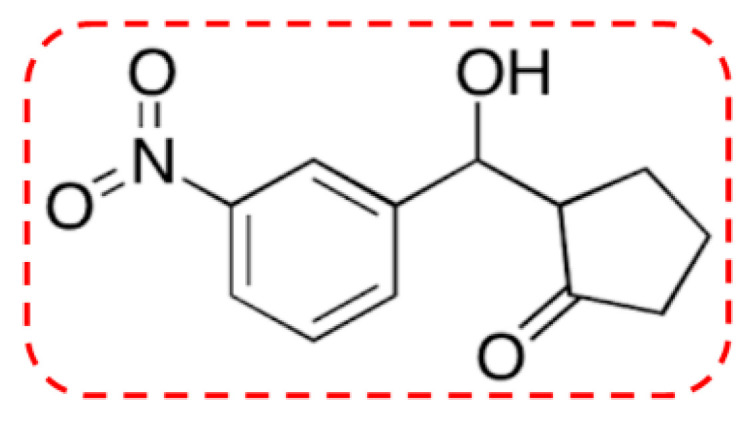
Chemical structure of 2-(hydroxy-(3-nitrophenyl)methyl)cyclopentanone.

**Figure 2 pharmaceuticals-13-00318-f002:**
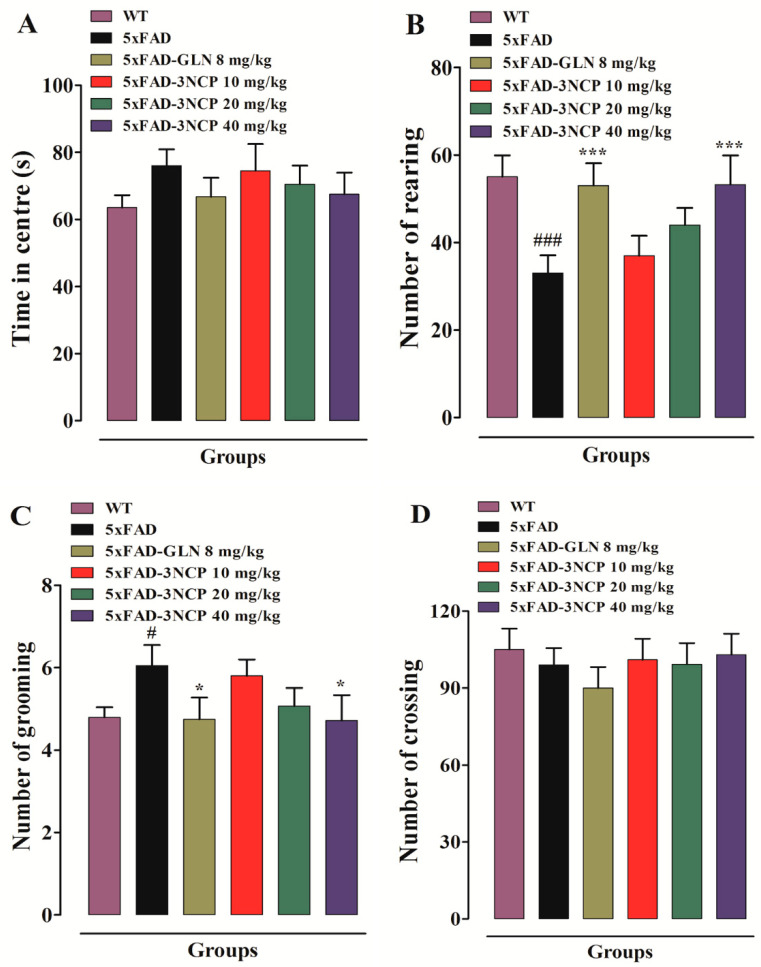
(**A**) Time in center; (**B**) number of rearing events; (**C**) number of grooming and (**D**) number of square crossing in the open field test after treatment with 3NCP/vehicle/galanthamine. Bars represent mean ± SEM. * *p* < 0.05, *** *p* < 0.001 compared to 5xFAD mice, ^#^
*p* < 0.05 and ^###^
*p* < 0.001 compared to non-transgenic WT-mice. Data was analyzed with one-way ANOVA followed by Dunnett post hoc test (*n* = 10 mice/group).

**Figure 3 pharmaceuticals-13-00318-f003:**
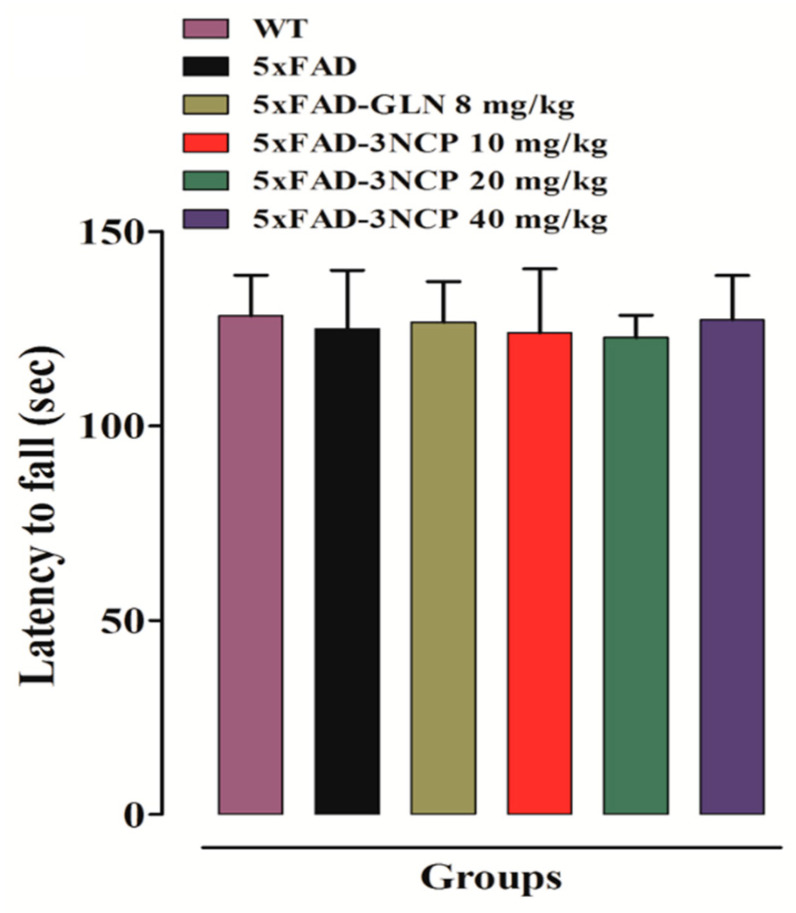
The effect of 3NCP on balance and motor coordination in the rotarod test. Bars represent mean ± SEM. Data was analyzed with one-way ANOVA followed by Dunnett post hoc test (*n* = 10 mice/group).

**Figure 4 pharmaceuticals-13-00318-f004:**
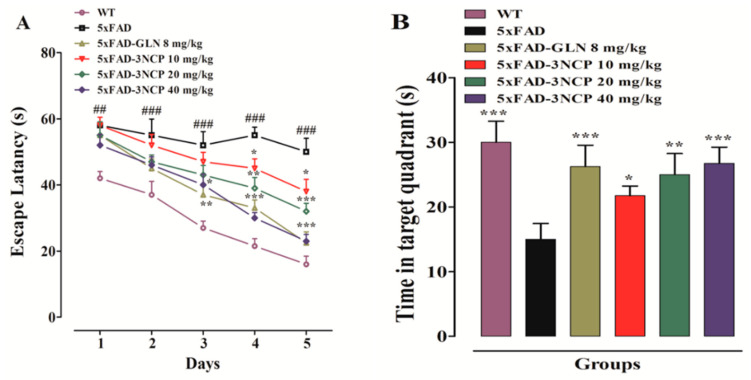
The effect of 3NCP on escape latency in MWM (**A**), time in target quadrant in probe test of MWM task (**B**), number of target crossing in probe test (**C**), and the effect of 3NCP on SAB in the Y-maze test (**D**). Bars express mean ± SEM. * *p* < 0.05, ** *p* < 0.01, *** *p* < 0.001 compared to 5xFAD group, ^##^
*p* < 0.001 and ^###^
*p* < 0.001 compared to non-transgenic WT group. Data was analyzed by two-way ANOVA followed by Bonferroni test, and one-way ANOVA followed by Tukey’s post-hoc test (*n* = 10 mice/group).

**Figure 5 pharmaceuticals-13-00318-f005:**
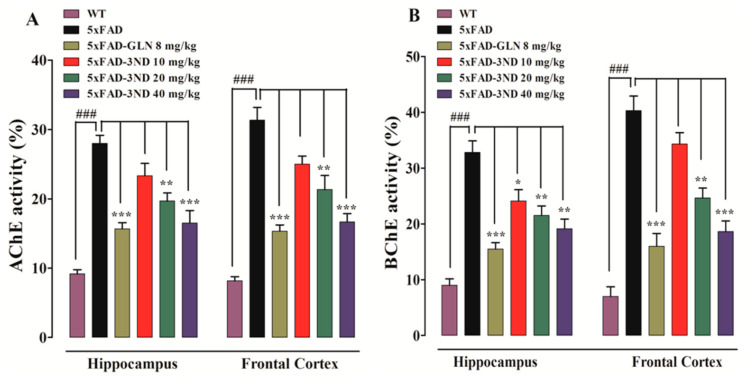
The effect of 3NCP on *AChE* level in HC and FC (**A**), the effect of 3NCP on the *BChE* level in both regions of brain (**B**). Data represent mean ± SEM. * *p* < 0.05, ** *p* < 0.01, *** *p* < 0.001 compared to 5xFAD mice, ^###^
*p* < 0.001 compared to non-transgenic WT-mice, one-way ANOVA followed by Tukey’s post hoc test (*n* = 4).

**Figure 6 pharmaceuticals-13-00318-f006:**
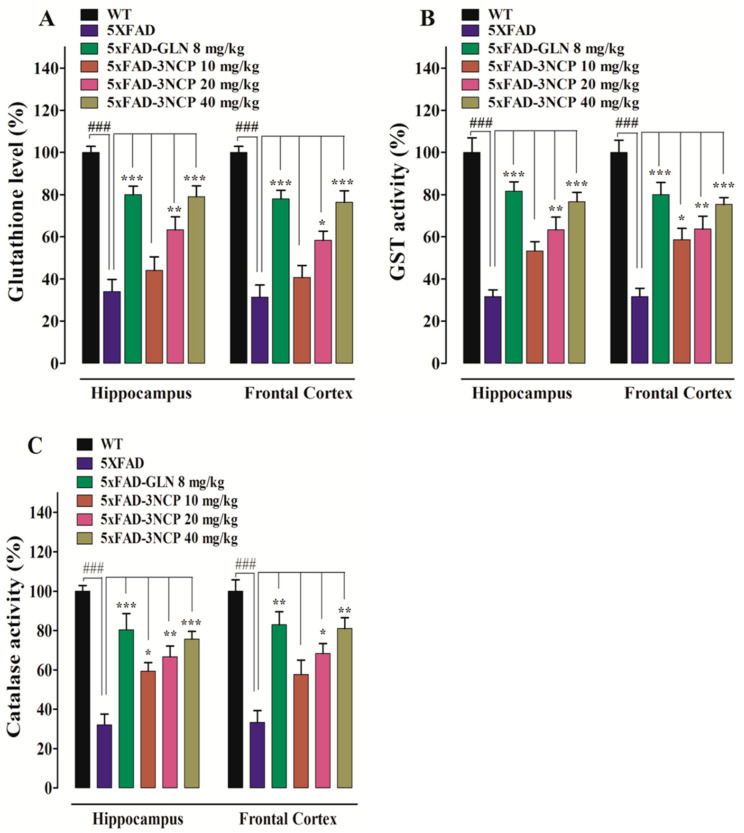
The effect of the 3NCP on %-level of GSH (**A**), %-activity of GST (**B**), and catalase (**C**), in the FC and HC. Bars displays mean level/activities (%) ± SEM.* *p* < 0.05, ** *p* < 0.01, *** *p* < 0.001 compared to 5xFAD group and ^###^
*p* < 0.001 compared to non-transgenic WT-group, one-way ANOVA followed by Tukey’s post hoc test (*n* = 4).

**Figure 7 pharmaceuticals-13-00318-f007:**
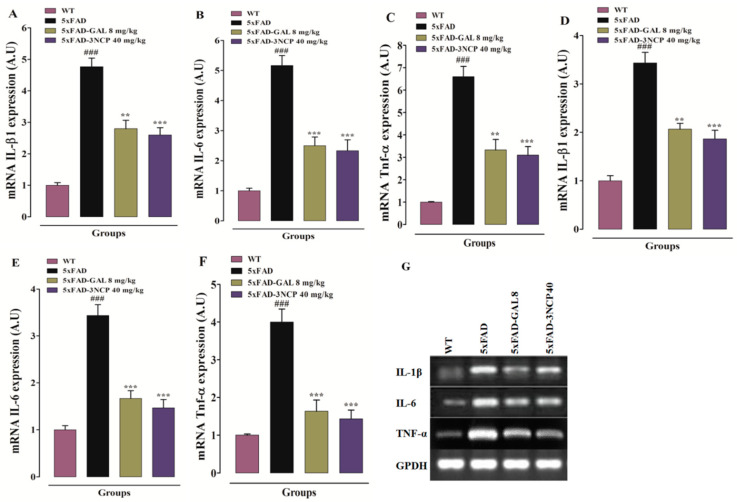
Animals were treated with 3NCP for 28 days and the mRNA expression of IL-1β, IL-6, and TNF-α in hippocampus (**A**–**C**) and frontal cortex (**D**–**F**) was measured. The effect of 3NCP on mRNA expression of IL-1β, IL-6, and TNF-α was confirmed by RT-PCR (**G**) The results were determined using ImageJ software and expressed in arbitrary unit (A.U). Bars indicate mean expression in A.U ± SEM. ** *p* < 0.01, *** *p* < 0.001 compared to 5xFAD-mice and ^###^
*p* < 0.001 compared to WT-mice (*n* = 4).

**Figure 8 pharmaceuticals-13-00318-f008:**
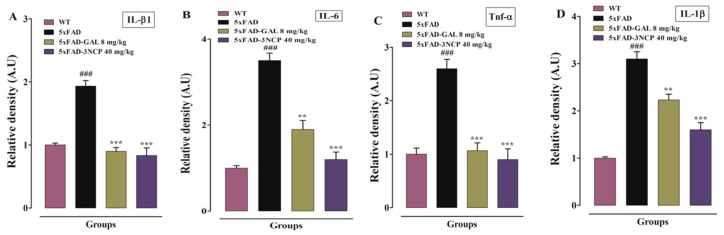
The relative density of IL-β1 (**A**), IL-6 (**B**), TNF-α (**C**) in FC and relative density of IL-β1 (**D**), IL-6 (**E**), and TNF-α (**F**) in HC of 5xFAD mice following treatment with 3NCP. Immunoblot of IL-β1, IL-6, and TNF-α (**G**). β-Actin was used as the loading control. The results were determined using ImageJ software and expressed in arbitrary unit (A.U). Bars indicate mean in A.U ± SEM. ** *p* < 0.01, *** *p* < 0.001 compared to 5xFAD-mice and ^###^
*p* < 0.001 compared to WT-mice (*n* = 4).

**Table 1 pharmaceuticals-13-00318-t001:** Results of in vitro cholinesterases inhibitory assay of 2-(hydroxy-(3-nitrophenyl)methyl)cyclopentanone.

Samples	Conc. (µg/mL)	Acetylcholinesterase (*AChE*)		Buterylcholinesterase (*BChE*)	
		Inhibition (%)	IC_50_ µg/mL	Inhibition (%)	IC_50_ µg/mL
3NCP	07.8 15.6	16.17 ± 1.53 25.0 ± 5.0 *	16.17	66.0 ± 2.65 224.0 ± 5.29	20.51
	31.25	45.7 ± 4.04		44.3 ± 6.03	
	62.5	57.0 ± 3.0		55.0 ± 4.36 *	
	125	65.3 ± 4.16		65.8 ± 1.89	
	250	72.3 ± 3.06		71.0 ± 5.57	
	500	77.0 ± 3.0		76.0 ± 1.73	
	1000	84.9 ± 5.0		86.3 ± 5.51	
Galantamine	07.8 15.6	11.3 ± 1.6 36.0 ± 2.0	13.12	16.0 ± 2.0 31.7 ± 5.77	14.43
	31.25	54.0 ± 3.0		57.0 ± 2.0	
	62.5	62.0 ± 2.0		70.0 ± 5.0	
	125	71.0 ± 2.2		75.0 ± 5.0	
	250	77.0 ± 3.0		80.0 ± 4.5	
	500	84.7 ± 2.0		88.0 ± 4.0	
	1000	88.7 ± 4.15		91.3 ± 2.31	

It has concentration-dependent inhibitory properties against these enzymes. Inhibition was observed as 16.17% and 6% at 7.8 µg/mL, while it was 84.9% and 86.3% at 1000 µg/mL. The IC_50_ values of 16.17 and 20.51 µg/mL against *AChE* and *BChE* were determined. Similarly, galanthamine caused inhibition of *AChE* and *BChE*, which was 11.3 and 16% at 7.8 µg/mL, whereas it was 88.7% and 91.3% at 1000 µg/mL, with IC_50_ values of 13.12 and 14.43 µg/mL. * *p* < 0.005 compared to galanthamine treatment.

**Table 2 pharmaceuticals-13-00318-t002:** Primers sequences for PCR reaction.

Primer	Sequence (5′-3′)
Internal positive control	CTAGGCCACAGAATTGAAAGATCT
	GTAGGTGGAAATTCTAGCATCATCC
Transgene	AGGACTGACCACTCGACCAG
	CGGGGGTCTAGTTCTGCAT
IL-1β	AGAAGCTTCCACCAATACTC
	AGCACCTAGTTGTAAGGAAG
IL-6	GCCCTTCAGGAACAGCTATGA
	TGTCAACAACATCAGTCCCAAGA
TNF-α	CTTCTCCTTCCTGATCGTGG
	GCTGGTTATCTCTCAGCTCCA
GAPDH	TGCACCACCAACTGCTTAGC
	GGCATGGACTGTGGTCATGAG

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
