# Peer review of "Attenuation of Spatial Memory in 5xFAD Mice by Halting Cholinesterases, Oxidative Stress and Neuroinflammation Using a Cyclopentanone Derivative"

_pharmaceuticals, 2020, doi:10.3390/ph13100318_

Round 1
Reviewer 1 Report
Introduction
Line 83.85: odd sentence, bad formulation, please check again!
Line 109-110. This postulation needs a reference!
Results and M&M
Figure 4: the figure legend is not informative. Confusing related to figure 4A, and the labeling of “A” is missing.
The figure legend can be improved. Better description is needed.
Line 204: please describe the abbreviation of the MWM test in the 2.3.3. title!
Comments: I think it is better that the order of the results be changed. Usually it is more logic that animal behavior is described at the beginning of the study followed by pathological or in vitro vs. ex vivo studies.
I wonder why the authors did chose 5x FAD mice model. This is not an optimal animal model for the AD since there are no patients that carrying both mutations, e.g. presenilin and APP, or tau.
Besides, I believe that the authors should describe better these AD-like mice model in the 4.2 Animal sections. What is the significance and noteworthy of this mice model for this study?
I wonder why the authors did not study the pathological changes in the 5X FAD animals, controls, and treated mice involved in the study. I would like to see the pathological changes after treatment with 3NCP by immunohistochemical staining. This can improve the paper tremendously. It would be interesting to see if the ex vivo data follows by the pathological changes.
Does the anti-inflammatory 3NCP change the morphology of the neurons in the hippocampus and frontal cortex?
It would be interesting to complete the RT-PCR data with protein expression examination in the brain tissues (here hippocampus and frontal cortex). This can be either shown by western blotting or by immunohistochemically technique. the RT-PCR data may not reflect the protein expression of these cytokines in the brain tissue.
Discussion and abstract
The discussion part is relaying mostly on the behavioral tests and less correlation of these results to the in vitro and ex vivo data. Reversely, the effort in the abstract was on the in vitro and ex vivo results and lacking the behavioral results.
Reviewer 2 Report
The results of the research presented in the manuscript add value to the on-going worldwide search for efficient treatment of Alzheimer’s disease. The manuscript represents a well-planned research experiment. However, there are some revisions needed before the manuscript can be accepted for publishing:
- In my opinion, the word “analogue” should be replaced with “derivative” in the title and elsewhere in the text. Analogue means a substitute with the same/similar properties, whereas 2-(hydroxyl-(3-nitrophenyl)methyl)cyclopentanone is a derivative of cyclopentanone.
- The name of this compound should be checked in the text and all gaps should be erased in it.
- The authors should provide more information regarding the reason for choosing this compound as an object of their research.
- In vivo, ex vivo, in vitro should be in italics throughout the text.
- Table 1, Table 2 in the text should start with the capital letter T
- Table 1: „Inhibitions“ should be replaced with „Inhibition“.
- Captions of Fig. 2, 3, 4, etc. should be rewritten not to start with the words „shows“ or „depicts“. For example, Fig. 2. „The effect of 3NCP on AChE level in HC and FC,...“. The style of the captions of different Figures should be checked and made uniform in respect of indicating (A), (B), etc. (place in the text, normal/bald letters and brackets)
- Line 110: „nitrophenyl“
- Line 112: „Ketone body“ should be replaced by „Ketone moiety“.
- Lines 248-249 and 252: the repeating information should be removed.
- Line 340: dipotassium hydrogen phosphate?
- Line 341: the EDTA name should be corrected, extra gaps should be erased.
- Line 362: 5,5’-Dithiobis (2-nitrobenzoic acid)? The missing bracket should be added as well.
- Line 449: 8 min.
- Reference list must be corrected to adhere to the formatting requirements of the journal.
- The authors should carefully check the text and insert gaps between words, numbers, and sentences where needed.
There are numerous English language mistakes to be corrected, for example:
- Lines 48-49: „In connection with this study“ should be replaced by, for example, „In this study“ or similar phrase as the described investigation was carried out in the frame of the reported research and not aside.
- Lines 83-87; 100-103; 134 (%); 184-187; 225-227; 252-255; 263 (of); 342 (were is missing); 422-424; 444-445 (were measured): sentences should be corrected to be clear and grammatically correct.
- Line 169: homogenated?
- Line 197: comma is not needed.
- Line 397: „are shown in table.“ should be erased.
Round 2
Reviewer 1 Report
The changes of the research presented in the manuscript after suggestion by reviewer is satisfactory.
However, there still are som mistakes when you wrote IL-1beta. USe the correct symbol for it in the figure legend 8, and som spaces between words can be adjusted.
Please add molecular weight of the antibodies and markers that you used in the western blot analyses.
Author Response
Response to Reviewer 1 Comments
Point 1: The changes of the research presented in the manuscript after suggestion by reviewer is satisfactory. However, there still are som mistakes when you wrote IL-1beta. USe the correct symbol for it in the figure legend 8, and som spaces between words can be adjusted.
Response 1: Many thanks for appreciation, valuable suggestions and comments. Changes have been made as per comments of the reviewer. We have thoroughly revised the manuscript for any typo mistakes and add symbol of IL-1beta “IL-1β” (Page# 18, Line 261-263).
Point 2: Please add molecular weight of the antibodies and markers that you used in the western blot analyses.
Response 2: Thanks for comment. The molecular weight of the antibodies and marker has been added in the revised manuscript as per reviewer kind suggestion (Page# 31, Line 520-521).
"Please see the attachment."